# Hyperbolic Representation Learning for Spatial Biology: Evaluating Cell Type Hierarchies in Breast Cancer Imaging Data

Youssef Wally[1], Johan Mylius-Kroken[1], Michael Kampffmeyer[1,2], Rezvan Ehsani[3], Vladan Milosevic[3], and Elisabeth Wetzer[1]

[1]Department of Physics and Technology, UiT The Arctic University of Norway, Tromsø, Norway
[2]Norwegian Computing Center, Oslo, Norway
[3]Department of Clinical Medicine, University of Bergen, Bergen, Norway
[1]{youssef.m.wally,johan.m.kroken,elisabeth.wetzer,michael.c.kampffmeyer}@uit.no
{v.milosevic,rezvan.ehsani}@uib.no

## Abstract

We demonstrate that hyperbolic representation learning effectively captures hierarchical cellular relationships in breast cancer. Using information-theoretic metrics, Lorentzian embeddings are shown to preserve significantly more biologically meaningful structure than Euclidean ones. Code: https://github.com/youssefwally/FlatlandandBeyond.

## 1 Introduction

Encoding hierarchical structure is a central challenge in representation learning. Hyperbolic models, operating in negatively curved spaces, naturally capture such hierarchies and often outperform Euclidean embeddings across domains including language, vision, and knowledge graphs [1–3].

In biological data, especially from Imaging Mass Cytometry (IMC), cells exhibit hierarchical relationships across types and states, captured through dozens of protein markers at subcellular resolution [4]. Modeling these relationships requires geometry-aware representations, an area where hyperbolic embeddings show strong potential [5]. Yet, most studies rely on qualitative visualization rather than rigorous quantitative evaluation.

We address this gap with an information-theoretic, geometry-agnostic framework for clustering evaluation based on mutual information (MI), conditional mutual information (CMI) using the Kraskov–Stögbauer–Grassberge (KSG) estimator [6]. Applied to a 42-marker breast cancer IMC dataset, we show that Lorentzian embeddings preserve substantially more biologically meaningful structure than Euclidean ones, and we release open-source tools for Lorentzian MI estimation and hyperbolic UMAP visualization.

## 2 Methodology

Traditional quantitative metrics such as the Silhouette Score or Average Distortion Index [7, 8] assume Euclidean geometry; linear distances, convex neighborhoods, and isotropy. These assumptions fail in hyperbolic spaces, where distances grow exponentially, and local curvature which affects neighborhood structure. Even substituting Euclidean distances with geodesics can yield misleading results due to the indefinite nature of the Lorentzian inner product and curvature-dependent spread.

Similarly, visualization methods like t-SNE and UMAP [9, 10] exhibit bias towards Euclidean geometry. Thus, assessing which geometry better captures biologically meaningful structure requires evaluation methods that do not assume Euclidean geometry.

We adopt a non-parametric MI estimator based on $k$-nearest neighbor (kNN), specifically the KSG estimator [6], which can be utilized to operate on arbitrary metric spaces, including Lorentzian geodesics.

**Geometry-Agnostic:** MI and CMI can be estimated directly from pairwise distances, independent of curvature, convexity, or coordinate representation [11]. This allows fair comparison between embeddings learned in Euclidean and hyperbolic spaces.

**Local and Density-Aware:** Unlike global clustering scores, kNN-based MI captures local density variations and neighborhood consistency.

**Cross-Geometry Alignment:** By estimating $I(X;Y)$ (MI), where $X$ and $Y$ denote Euclidean and hyperbolic representations respectively, we quantify the shared information between representations, providing a direct measure of structural preservation.

### 2.1 KSG Estimator Formulation

Given random variables $X$, $Y$, and $Z$, the CMI under the KSG estimator can be expressed as

$$I(X;Y|Z) \approx \psi(k) + \psi(N)$$
$$-\frac{1}{N}\sum_{i=1}^{N}\left[\psi(n_x^{(i)}+1) + \psi(n_y^{(i)}+1) - \psi(n_z^{(i)}+1)\right] \quad (1)$$

where $\psi(\cdot)$ is the digamma function, and $n_x^{(i)}, n_y^{(i)}, n_z^{(i)}$ denote the number of neighbors within

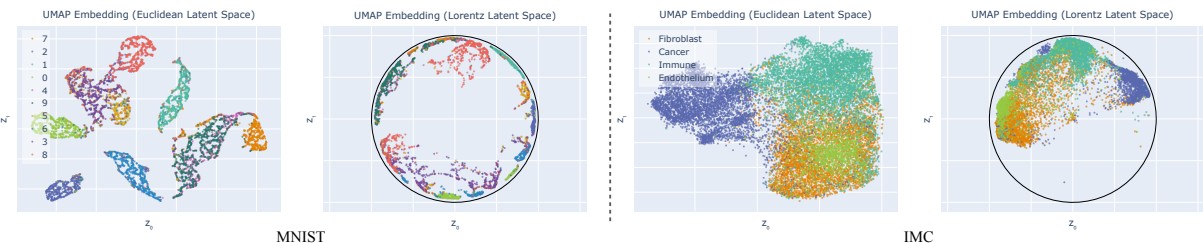

**Figure 1.** Embeddings in 2D latent space of VAEs. Colors represent ground truth labels.

the $\varepsilon_i$-ball of the corresponding variables, excluding the query point. The radius $\varepsilon_i$ is defined as the maximum distance to the $k$-th nearest neighbor in the joint space. The mutual information (MI) case follows directly by omitting the (Z)-dependent term.

# 3 Data

We use the IMC dataset from [5], featuring a 42-marker panel for phenotypic and spatial profiling of the tumor microenvironment, with emphasis on cancer-associated fibroblasts in breast cancer. Hierarchical cell annotations span four levels; we use the first three, from broad categories (Cancer, Immune, Endothelial, Fibroblasts) to fine-grained immune subtypes and detailed T cell and macrophage phenotypes. We also benchmark our method on the MNIST handwritten digit dataset [12].

# 4 Experiments

## 4.1 Implementation Details

Experiments were conducted in PyTorch [13] using Riemannian optimization [14] via Geoopt [15], with 32-bit precision as in [16]. Models include Hyperbolic Variational Autoencoder (HVAE), and Euclidean Variational Autoencoder (EVAE). **All analyses are performed on the test set.** To ensure fair comparison, H-VAE and E-VAE are trained independently, with reconstruction loss as a common objective.

# 5 Results

## 5.1 Qualitative Analysis

Visualizations of Euclidean and Lorentzian embeddings (Fig. 1) reveal clear structural differences that highlight the representational advantages of hyperbolic geometry. In Lorentzian space, clusters appear more compact and hierarchically organized, consistent with the space's exponential volume growth. In the IMC dataset, minority classes such as Endothelial Cells (8.40% of total samples) form tighter, more separable clusters than in Euclidean space.

**Table 1.** Estimated MI and CMI on IMC and MNIST test sets.

| Quantity | IMC | MNIST |
|---|---|---|
| $MI(D_L; C)$ | **1.07** | **1.86** |
| $MI(D_E; C)$ | 0.96 | 1.78 |
| $MI(D_L; D_E)$ | 0.01 | 4.03 |
| $CMI(D_L; C \mid D_E)$ | **1.06** | **0.16** |
| $CMI(D_E; C \mid D_L)$ | 0.00 | 0.09 |

This indicates that Lorentzian embeddings capture fine-grained biological distinctions even among underrepresented cell types.

Similar behavior is observed in MNIST, where ambiguous digits such as certain "3"s are positioned between clusters of visually similar digits ("0", "6", "8"), reflecting Lorentz space's ability to represent semantic uncertainty. In contrast, Euclidean embeddings enforce flatter separations that obscure such relationships.

## 5.2 Quantitative Analysis

We evaluate how well each geometry encodes class-relevant structure using MI between pairwise distance matrices Lorentzian Distances ($D_L$), Euclidean Distances ($D_E$) and class labels ($C$). We also compute CMI to quantify the incremental information each geometry contributes beyond the other.

The MI results confirm that Lorentzian embeddings encode more class-relevant information ($MI(D_L; C) > MI(D_E; C)$) in both datasets. The near-zero $MI(D_L; D_E)$ on IMC indicates that the two geometries capture largely non-overlapping structural information. Moreover, $CMI(D_L; C \mid D_E) = 1.06$ versus $CMI(D_E; C \mid D_L) = 0.00$ shows that Lorentzian geometry provides additional, non-redundant information beyond what Euclidean structure explains, demonstrating superior expressiveness and alignment with biological hierarchies.

# 6 Conclusions

We show that unsupervised hyperbolic representation learning more effectively captures the hierarchical structure of breast cancer cell relationships than its Euclidean counterpart.

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
