# OpenReview forum: "Hyperbolic Representation Learning for Spatial Biology: Evaluating Cell Type Hierarchies in Breast Cancer Imaging Data"
_NLDL.org/2026/Abstracts_Track — NLDL 2026 Abstracts_

### Official Review · Reviewer_VZ3k · 2025-10-30

**Soundness:** 4
**Correctness:** 4
**Rating:** 5
**Confidence:** 4

**Summary:**

This abstract presents a geometry-agnostic, information-theoretic framework for evaluating representations learned in hyperbolic versus Euclidean spaces, applied to imaging mass cytometry (IMC) data from breast cancer tissues. Using mutual information (MI) and conditional mutual information (CMI) estimated via the KSG estimator, the authors show that Lorentzian (hyperbolic) embeddings preserve biologically meaningful hierarchical relationships between cell types more effectively than Euclidean embeddings. The study employs Hyperbolic and Euclidean Variational Autoencoders (H-VAE and E-VAE) and releases open-source code for MI estimation and visualization.

**Strengths:**

Novel evaluation framework:
The proposed geometry-agnostic MI/CMI approach is a strong conceptual contribution. It avoids the Euclidean bias of common metrics (e.g., Silhouette Score, Distortion Index), enabling fair comparison of embeddings in different geometries.

Strong theoretical grounding:
The use of the KSG estimator and its adaptation to Lorentzian metrics is well-motivated and mathematically sound. The inclusion of explicit equations and clear links between theory and biological application enhance credibility.

Biological significance:
Demonstrating that hyperbolic embeddings better capture cell-type hierarchies in breast cancer is both biologically and computationally relevant, bridging machine learning and spatial biology.

Clarity and reproducibility:
The methodology and results are concisely described, with well-structured figures and a public code repository, supporting transparency and potential reuse.

Interdisciplinary value:
The work connects representation geometry with cellular spatial organization, a promising direction for future multimodal bioinformatics research.

**Weaknesses:**

Limited experimental depth:
The study focuses on one IMC dataset and MNIST as a synthetic benchmark. Broader validation across more biological datasets (e.g., spatial transcriptomics or different cancer types) would strengthen generality.

Abstract-level presentation:
The abstract lacks explicit details on embedding dimensionality, hyperparameters, and training stability, which are important for evaluating the robustness of Lorentzian VAEs.

Interpretability and biological insight:
While MI-based metrics quantify structure preservation, biological interpretation could be enriched — e.g., which specific immune subtypes or fibroblast phenotypes are better separated in hyperbolic space.

Comparative baselines:
Results would benefit from inclusion of other non-Euclidean approaches (e.g., Poincaré embeddings, spherical embeddings) to contextualize Lorentzian advantages.

---

### Official Review · Reviewer_tsak · 2025-10-31

**Soundness:** 3
**Correctness:** 3
**Rating:** 4
**Confidence:** 4

**Summary:**

The abstract explores hyperbolic representation learning by comparing Lorentzian embeddings to Euclidean embeddings within VAE latent spaces.

The authors evaluate the effectiveness of hyperbolic representations through both qualitative and quantitative analyses.

**Strengths:**

•	Well-written and clearly introduces the problem and its relevance.

•	Demonstrates results using both quantitative and qualitative evaluation methods.

**Weaknesses:**

•	The submission includes an incorrect submission ID.

•	The authors could clarify their novel contributions more explicitly i.e. what aspects of their approach go beyond prior work?

---

### Official Review · Reviewer_Kgpi · 2025-11-03

**Soundness:** 4
**Correctness:** 4
**Rating:** 5
**Confidence:** 4

**Summary:**

The abstract shows that the use of hyperbolic representations, more specifically Lorentzian embeddings, capture biologically meaningful structures better than Euclidean ones. This is measured by using information-theoretic metrics such as mutual information and conditional mutual information.

**Strengths:**

The abstract is nicely written and easy to follow, and includes intuitive visualisations that show the contribution of the work. The quantitative analysis continues this and strengthens the contribution of the work

**Weaknesses:**

Overall a very interesting abstract, only a small note on that the text size of the titles in Figure 1 could be a bit larger.

---

### Decision · Program_Chairs · 2025-11-05

**Decision:**

Accept

**Comment:**

The abstract is of interest to the community and should be presented at the conference.